# Effect of a Dietary Essential Oil Blend in Dairy Cows during the Dry and Transition Period on Blood and Metabolic Parameters of Dams and Their Calves

**DOI:** 10.3390/ani14010150

**Published:** 2024-01-02

**Authors:** Cangir Uyarlar, Abdur Rahman, Eyup Eren Gultepe, Ibrahim Sadi Cetingul, Ismail Bayram

**Affiliations:** 1Department of Animal Nutrition and Nutritional Diseases, Faculty of Veterinary Medicine, Afyon Kocatepe University, ANS Campus, Afyonkarahisar 03000, Turkey; cangiruyarlar@hotmail.com (C.U.); eegultepe@gmail.com (E.E.G.); sadicet@yahoo.com (I.S.C.); ibayram@aku.edu.tr (I.B.); 2Department of Animal Sciences, University of Veterinary and Animal Sciences, Jhang Campus, Jhang 35200, Pakistan

**Keywords:** essential oil, dairy calves, phytogenic, dairy cows

## Abstract

**Simple Summary:**

The transition period is the most critical phase in which cows face a sever metabolic and hormonal stress. During this period; animal’s feed intake is reduced while its metabolic need of nutrients increases. This specific situation may develop immunosuppression and metabolic distress leading to the poor health status and production of the cow and calf. Farm mangers and nutritionists adopt various entanglement and nutritional strategies including herbal products; to reduce the severity of these issues. Essential oils are the organic products and known to enhance the feed intake; improve immune status and reduce metabolic problems in the dairy herd. This study revealed that addition of essential oils blend have improved the immune status and energy metabolism in dairy cows during transition period without putting any adverse effects on the calf health.

**Abstract:**

Cows face severe challenges of immunosuppression and negative energy balance during transition periods. The current study was designed to investigate the effects of essential oil blend supplementation in dairy cow’s ration during dry periods on the health of the cow and calf. In the current study, 45 dry Holstein dairy cows were divided into three groups, each consisting of 15 animals. The control group was without any supplementation; the positive control group was only injected with 0.2 mg/kg levamisole (intramuscular) at 2 months before parturition and 1 month before parturition; and the treatment group was supplemented with 3 g/day for each cow essential oil blend mixed in total mixed ration (TMR). A mixed vaccine against *E. coli*, Rotavirus, and Coronavirus was administered to all cows before 42 days and after 21 days of calving. The day of the calving of the animal was accepted as day 0. Blood samples were collected from the coccygeal vein of all cows at −60, −45, −30, −15, −7, 0, 1 and 2 days, and the colostrum samples were taken on days 0, 1, and 2. Blood samples were also collected from the jugular vein (V. Jugularis) of the newborn calves on days 0, 1, and 2. The results of the hematological parameters revealed no difference in the total red blood cell count, hemoglobin amount, hematocrit, MCV, MCH, MCHC, RDW, PLT, MPV, and PCT values in both prenatal and postnatal blood of the cows (*p* > 0.05). In terms of immunological parameters, the total IgG level was significantly higher in the experimental group on the 7th day before birth compared to the other two groups, while the total leukocyte count, lymphocyte count, monocyte count and granulocyte counts were found to be lower after birth (*p* < 0005). Blood NEFA and BHBA levels were also lower in the experimental group compared to the other two groups (*p* < 0.005), and the blood glucose level was not different (*p* > 0.05). In calves, no difference was found between the groups in any of the parameters examined (*p* > 0.05). It is concluded that the dietary addition of an essential oil blend during the transition period enhanced the immune status and energy metabolism of cows without any effect on the health status of newborn calves.

## 1. Introduction

The transition period comprises the last three weeks of gestation (close-up dry period) and the first three weeks of lactation (fresh period) [1]. Studies on the nutrition and physiology of transition dairy cows have shown that the transition period is the most crucial period for herds. Metabolic diseases in this period cause great economic losses [2]. The requirements of glucose, fatty acids and minerals are rapidly changing during this period. It has been reported that the management of dairy cows during the transition period is critical for maintaining the maximum milk yield to avoid economic losses at the farm and preventing metabolic diseases [2,3]. The main reasons for this challenge are a decrease in feed consumption as the birth approaches, rumen shrinkage, and a significant decline in its capacity due to the developing volume of the fetus. As the milk synthesis commences even at the pre-calving stage and experiences a rapid increase after calving, the demand for nutrients surges exponentially. However, feed consumption does not keep pace with this increase, primarily due to diminishing rumen capacity and hormonal stresses. Thus, the nutrients stored in the body are quickly consumed. In particular, metabolism puts the animal into a negative energy balance by encountering difficulties in bridging the energy gap. In addition, during the transition period, dairy cows undergo a significant change in feed quality and feeding pattern, which shifts from a low-cellulose diet during the dry period to a high-energy ration following calving [3]. Therefore, to maintain maximum milk yield and to protect cows from metabolic diseases during the transition period, the management of nutrient intake such as glucose, fatty acids, and minerals is highly important [2]. The need for glucose triples, amino acids double, and fatty acids increase fivefold compared to the requirements on the 250th day of pregnancy and the 4th day of lactation [4].

Many aromatic plants are used in various fields due to the active chemical compounds found in their seeds, fruits, leaves, or roots, and they have different modes of action. In terms of animal nutrition, these plants have appetizing and digestive-stimulating properties, as well as antiseptic effects. Although, the effects vary according to the active ingredients, i.e., many essential oils have antimicrobial, carminative (herbal mixtures that prevent gas accumulation in GIT), choleretic (liver booster for bile secretion), sedative, diuretic, and antispasmodic effects [5,6]. All essential oils strengthen the immune system by increasing the production of IgG and IgA [7,8]. It has also been stated that optimum rumen fermentation can be achieved by suppressing some rumen microorganisms with herbal extracts [9]. It was determined that the addition of mint (5%) to cattle rations in early lactation did not affect dry matter consumption, but increased feed consumption time (minute/kg DM) and decreased ruminal pH [10].In studies conducted in pigs, the addition of thyme oil to the ration during weaning stress supported the immune system and abolished the effects of stress-related immunosuppression [11]. Frankic et al. [12] showed that plant extract supplementation in pigs reduced DNA damage in lymphocytes. This demonstrated the potential beneficial effects of plant extracts on the immune systems of animals exposed to diet-induced oxidative stress. In another study by Sads and Bilkei [13], a lower incidence of disease was found in weaning pigs fed with a 1000 ppm thyme supplement compared to the animals with no supplementation. Michiels et al. [14] also reported a decrease in the number of intra-epithelial lymphocytes in the distal small intestine of animals treated with 500 ppm carvacrol and thymol.

There are various individual or blends of essential oils, including Digestarom^TM^, that are being used in animal diets for the described purposes. Digestarom^TM^ is a commercially available registered product of a renowned company (DSM, The Netherlands). It is mainly a phytogenic blend of essential oils. Its main mode of action is anti-inflammatory, antioxidant, antimicrobial and immune-supportive. With this backdrop, the current study was designed to investigate the effects of the dietary supplementation of essential oil blend “Digestarom^TM^” (having variable quantities and types of herbal plant essential oils i.e., cinnamon, cumin, mint, thyme, garlic, rosemary; carrier substances: SiO_2_, NaCl), during the transition period on some biochemical and metabolic parameters of calves and their dams.

## 2. Materials and Methods

All experimental procedures used in this study were approved by the Animal Ethics Committee of the Afyon Kocatepe University (Case No.49533702/66).

A total of 45 Holstein dairy cows, which had finished their second lactation, with nearly the same body weights (550–650), body condition scores (3.5–4), and average milk yield in the last lactation (25–29 L/day), at a modern commercial dairy farm were selected in this trial and randomly divided into 3 groups consisting of 15 animals in each. There was a control group without any supplementation, and the positive control group was administered levamisole (0.2 mg/kg, intramuscularly) at the beginning of the study (02 months before parturition) and 1 month before parturition. The treatment group was supplemented with Digestarom^TM^ Dairy^®^ at a dose of 3 g/day per animal in the TMR of the animals throughout the study. The study period started 2 months before parturition and lasted until the second day after parturition and Digestarom^TM^ Dairy^®^ was supplemented throughout the study period. Levamisole was routinely administered to the animals as a dewormer, and it enhances immunity. In our study, we intentionally administered levamisole to one group and labeled it as the positive control; one group without levamisole was the control group and one with Digestarom^TM^ was set to differentiate the effects of levamisole and Digestarom^TM^.

### 2.1. Feeding Method and Sampling

Cows were fed Total Mixed Ration (TMR) throughout the trial period. The animals were fed twice a day, at 06:00 in the morning and 18:00. The TMRs of all groups were prepared as isocaloric and isonitrogenous. Group feeding was given to the animals. Groups were separated from each other and their feeds were weighed, mixed and given in separate compartments. Weende analysis (dry matter, crude ash, crude protein, crude fiber, crude fat) and Van Soest (ADF and NDF) analysis were performed according to the method suggested by Van Soest (1978), of the feed samples taken from the raw materials used in TMR. The ingredients and chemical compositions of the rations are given in Table 1.

### 2.2. Vaccination

All cows were vaccinated against *E. coli*, Rotavirus, and Coronavirus 42 days before parturition, and these were repeated 21 days before parturition.

## 3. Blood Sampling

Blood was drawn from the tail vein (v. coccygea) of all cows on days −60, −45, −30, −15, −7, 0, 1, and 2. The calving day was set as “0 (zero)”. Blood samples were drawn in tubes coated with EDTA (anticoagulant coated agent) for hematological parameter analysis and additional blood samples were placed in the tubes without anticoagulant. Samples without anticoagulant were centrifuged at 1500× *g* for 10 min and serum was separated and stored at −30 °C until analysis. For blood serological analysis, a portion of the blood samples collected at −60, −45, −30, −15, −7 and 0 day was used. For blood biochemical parameters analysis, blood samples were collected a day before parturition and a day after parturition. Similarly blood samples were also collected from calves on the second day of life for blood biochemical and hematological parameter analysis. The collected blood samples were analyzed for hematological parameters (total leukocyte count, TLC; lymphocyte count, LC; neutrophil count, NC; monocyte count, MC; red blood cell count, RBC; hemoglobin, Hb; hematocrit, HCT; mean corpuscular volume, MCV; mean corpuscular hemoglobin, MCH; mean corpuscular hemoglobin concentration, MCHC; platelet, PLT; mean platelet volume, MPV; mean platelet volume, PDW; platelet distribution width, PCT; platelet percentage was determined by a total blood analyzer (BC 2800 Vet, Mindray Medical International Ltd., Shenzhen, China).

In serum and plasma samples, alanine aminotransferase (ALT), aspartate aminotransferase (AST), alkaline phosphatase (ALP), gamma glutamyl transferase (GGT), nonesterified fatty acid (NEFA), beta hydroxy butyric acid (BHBA), glucose, and total immunoglobulin G (IgG) levels were determined using the relevant commercial kits in a fully automated ELISA reader device (Chemwell 2910).

## 4. Data Analysis

The Kolmogorov–Smirnov test was applied to obtain information on whether the data obtained in the study showed normal distributions for each parameter and time interval. Logarithmic transformation was applied to data that did not show normal distribution. If an outcome variable was not normally distributed, the natural logarithmic transformation was used for the scales. In brief, all observations underwent a logarithm transformation to base 10. Then, the log-transformed data were used for data analysis. Besides data analysis, for easy reading and further comparisons between the current data, the means were presented as is (natural numbers). Normal distribution was obtained in all of the data with logarithmic transformation. After this stage, the prenatal period was evaluated in separate blocks, together with the parturition data. In determining the differences between each parameter in the prenatal block, a Linear Model was created and one-way ANOVA was applied for repeated measurements. A one-way ANOVA test was used in the evaluation of parturition parameters and milk parameters, Bonferroni tests were applied as posthoc for parameters where the variances were equal, and Tamhane’s T2 tests were used for parameters where the variances were not equal in determining which groups caused the differences. A paired T test was used to evaluate after-parturition data. The significance level was set as *p* < 0.05. The values in the Table 2, Table 3, Table 4 and Table 5 are expressed as mean ± standard error. All calculations regarding the data obtained from the analysis were made using the MEDCALC^®^ program (MedCalc Statistical Software version 16.4.1, Oostende, Belgium).

## 5. Results and Discussion

There was no significant difference between the groups and among the groups regarding hemoglobin, percentage of hematocrit, mean erythrocyte volume, mean corpuscular hemoglobin amount, mean corpuscular hemoglobin concentration (MCHC), mean platelet volume and mean total red blood cell count before parturition, at parturition and after parturition stages, but there was in platelets count, which showed a lower value (*p* = 0.009) at day 60 as compared to 45, 30, 15 and 7 days before parturition among all groups with respect to time (Table 2). It is inevitable for the cows to develop immunosuppression due to metabolic and immunological stress over a period of approximately 60 days, including the period before and after parturition. This situation causes a critical period in terms of both maternal and calf health [15,16]. Suppression of the immune system paves the way for many postnatal infectious diseases that can endanger the life of the cow and the calf [17]. Kumar and Pachauri [18] reported that blood hemoglobin levels decreased during the dry period of cows fed pasture. Contrary to this study, Baqi and Rahman [19] and Rajora and Pachauri [20] reported that the blood hemoglobin level increased as the gestation period progressed. Shaffer et al. [21] reported that the total number of red blood cells increased during the winter season, while Kumar and Pachauri [18] reported that PCV, MCH, and MCHC values increased. Nazifi et al. [22] reported that the level of HT decreases after parturition in cows. The same researchers reported that prenatal red blood cell count (RBC) is higher than after parturition. The changes stated in the presented study were not encountered. The reason for this may be the barn conditions in which the animals were housed, with a more controlled and smaller number of animals during the experiment, the careful handling of daily nutrient needs by TMR feeding, and the limiting of environmental stress factors during the parturition period in our study as compared to previous reports in the literature. Similarly, no significant difference was observed in the hematology results (HE, HT, MCV, MCH, MCHC, PLT, MPV, and RBC) values of calves after birth (Table 5). The number of studies examining hematological parameters and the factors affecting this in calves is quite limited. Generally, studies have focused on comparing the situation between calves and adult cattle [23,24].

In the prenatal phase of the study, there was no significant difference between the groups in terms of total leukocyte count (TLC), lymphocyte count (LC), neutrophil count (NC), and monocyte count (MC) in the cows before parturition (Table 3). On the other hand, significant gradual increases were observed in the amount of TLC (*p* = 0.028), the amount of LC (*p* = 0.041), the amount of NC (*p* = 0.045), and the MC (*p* = 0.001) among the groups before parturition with respect to time (Table 3). However, a significantly higher value was found for the amount of Total IgG (*p* = 0.001) in the treatment group as compared to others before parturition (Table 3). Regarding the groups, the IgG level of the treatment group (T) supplemented with Digestarom^TM^ was found to be significantly higher than the other groups only on the 7th day before parturition. A significant increasing change was observed in IgG levels depending on time (*p* = 0.001) among the groups. The amounts of IgG, which were statistically similar to each other at 60 and 45 days before parturition, increased significantly on the 30th day, and this increase continued to represent a significant difference between the groups on the 7th day before parturition. After calving, a significant decrease in values was observed in the treatment group in the amount of TLC (*p* = 0.001), lymphocyte amount (*p* = 0.019), NC amount (*p* = 0.001), and MC amount (*p* = 0.042) as compared to the other groups. On the other hand, there was no significant difference in the amount of IgG between the groups (*p* = 0.112) after calving. There were no significant differences in TLC, LC, NC, MC, and IgG values according to the blood results of calves (Table 5). The amount of total IgG in the blood is one of the most important findings showing the immune status of an animal [25]. Accordingly, the IgG level increases as an immune response after vaccination and immune power [26,27]. In addition, the blood total leukocyte count increases rapidly on the day of parturition and after [28,29]. Merrill and Smith [28] showed a significant increase in lymphocyte and eosinophil counts and a decrease in neutrophil count up to ten days after parturition, and they also reported that there was no change in the monocyte count. Some researchers [30,31] reported that the mean blood total leukocyte and blood lymphocyte counts and neutrophil count decreased rapidly after birth, but increased again gradually until the 20th day of lactation. Saad et al. [32] reported that blood total lymphocyte count decreased before and after parturition, but increased again in the second week after parturition. In the presented study, the blood IgG level of the T group increased in the prenatal period, while TLS, LS, NC, and MC decreased in the postpartum period. On the other hand, the IgG level of the T group was numerically higher when compared to the control groups. This result indicates that Digestarom^TM^ application supports the immune system during the birth period in cows and has a protective effect against infectious diseases. Although the IgG level was significantly higher in the prenatal period in the treatment group, it became non-significant after parturition. The main reason for the disappearance of this statistical difference after parturition may be the individual differences in IgG levels transferred to the offspring with colostrum after birth.

There are not enough studies examining the effects of various feeding practices on cows before parturition on the health and development of newborn calves. Studies have mostly focused on periods of intense stress such as weaning or transport [26,27,33,34,35,36,37,38]. This is because immunosuppression develops with increased cortisol release during stress [39]. In the present study, no significant difference arose in the TLC, LC, NC, MC, or IgG values in the blood of calves when Digestarom^TM^ was given to cows in addition to the ration before parturition (Table 5).

Considering ALT, AST, ALP, GGT, NEFA, BHBA, and glucose values in blood serum, there was no significant difference between groups in the prenatal period (Table 4). On the other hand, the between-group analysis showed significantly lower values in the treatment group as compared to other groups in terms of serum NEFA (*p* = 0.001) and BHBA (*p* = 0.001) concentrations in cows after parturition. In addition, there was no significant difference between the groups in the blood serum ALT, AST, ALP, and GGT values of calves (Table 5). Serum AST, GGT, and ALT concentrations are indicators of hepatic functions in farm animals [40]. Sevinç et al. [41] reported that there is a relationship between serum AST and GGT levels, which are liver enzymes, and their abnormal values indicate fatty liver syndrome. In the presented study, no increase in any parameter of liver tests was observed before or after parturition in any group. NEFA and BHBA are blood parameters that provide important information about the metabolic diseases, especially fatty liver and ketosis during the transition period of cow. The fact is that if both NEFA and BHBA are high in the blood, it indicate that the animals have a high susceptibility to metabolic diseases [2,42,43,44]. Plasma NEFA and BHBA levels are a more reliable indicator of the energy metabolism and fat mobilization from body reserves as a result of negative energy balance [45]. In particular, the acute rise in NEFA at calving is associated with the onset of triglyceride (TG) infiltration [46,47]. In the presented study, Digestarom^TM^ administration to dairy cows before calving helped to reduce blood NEFA and BHBA levels after calving. This effect might be attributed to the increased feed intake, feed palatability, and its efficient digestion to ensure more nutrient availability for the body due to the addition of the essential oil blend, as it helps in improving the gut functions as well.

To the best of our knowledge, no study has been found that examines the effects of an application made as a supplement to the prenatal ration on the blood biochemistry parameters of newborn calves on the day of birth and in the following two days. In the present study, Digestarom^TM^ administered to cows before birth via a supplement to the ration did not cause significant differences in the blood ALT, AST, ALP, or GGT values at birth of newborn calves (Table 5).

## 6. Conclusions

In this study, the effect of a plant-derived essential oil mixture, called Digestarom^TM^, supplemented to dairy cows during the dry period on postpartum immunity and metabolic parameters in cow and calves was investigated. Supplementation of Digestarom^TM^ supported the immune systems of cows before and after birth, helped to prevent immunosuppression, and suppressed the rise of the most important parameters of energy metabolism, such as NEFA and BHBA, in cows after parturition. Thus, it has been observed that this application has beneficial effects on supporting both the immune system and energy metabolism in cows. Future extensive studies are needed to investigate the effect of Digestarom^TM^ administration on immune and metabolic parameters during lactation in dairy cows.

## Figures and Tables

**Table 1 animals-14-00150-t001:** Ingredients and chemical compositions of the rations (DM%).

Feedstuffs (¥ DM%)	Before Parturition	Early Lactation
Corn silage	16.7	19.7
Dry alfalfa	5.7	10.3
Wheat straw	34.7	0
Barley grain	5.8	8.5
Corn grain	11.2	17.2
Corn gluten	-	3.6
Whole cottonseed	-	7.8
Soybean meal (48%CP)	2.7	4.9
Lentil	3.6	2.9
Canola meal	2.4	2.3
DDGS *	6.1	5.7
Wet brewer’s grain	3.2	6.2
Wheat bran	5.8	4.3
SoyPass **	2.1	3.8
Fat, Ca soap	0	1.7
Molasses ***	0.3	2.8
Limestone	0.3	0.7
Salt	0.4	0.57
Vitamin–mineral mixture ****	0.04	0.04
MgS0_4_	0	0.8
Yeast	0	0.3
Toxin binder ^≠^	0.02	0.03
**Chemical composition of the rations (%DM)**		
Crude protein	12.6	17.7
RDP (% of CP)	73.6	61.3
Bypass Protein	26.4	38.7
NEL (Mcal/kg)	1.51	1.71
NDF	45.5	37.7
ADF	31.6	24.6
Ca	0.54	0.89
*p*	0.21	0.41

¥ dry matter; * dried distiller’s grain solubles; ** rumen protected protein; *** a liquid byproduct of sugar industry; **** each 1-ton premix contains vitamin A 15,000,000 IU, vitamin D 3,000,000 IU, vitamin E 75,000 mg, vitamin B1 15,000 mg, niacin 300,000 mg, biotin 2000 mg, manganese 50,000 mg, iron 50,000 mg, zinc 100,000 mg, copper 20,000 mg, cobalt 200 mg, iodine 800 mg, selenium 260 mg; ^≠^ commercially available product.

**Table 2 animals-14-00150-t002:** Hematological values of cows fed essential oil blends before and after parturition ($\bar{x}$ ± *S*$\bar{x}$).

**Blood Hemoglobin Levels of Cow**										
	Before Parturition					*p*	Parturition	After Parturition		*p*
**Groups**	−60	−45	−30	−15	−7	0.239	0. Day	1. Day	2. Day	
**Control**	11.23 ± 0.11	10.86 ± 0.10	10.74 ± 0.11	10.83 ± 0.08	10.67 ± 0.10		9.48 ± 0.31	9.62 ± 0.08	9.74 ± 0.12	0.621
**Positive control**	11.20 ± 0.09	10.95 ± 0.11	10.86 ± 0.10	10.71 ± 0.10	10.73 ± 0.12		9.63 ± 0.28	9.79 ± 0.08	9.57 ± 0.12	0.389
**Treatment**	10.90 ± 0.12	10.86 ± 0.11	10.77 ± 0.11	10.79 ± 0.11	10.87 ± 0.13		9.53 ± 0.19	9.72 ± 0.11	9.74 ± 0.10	0.863
** *p* **	0.625						0.112	0.472	0.273	
**Blood Hematocrit Levels of Cow**										
**Control**	28.24 ± 0.24	28.51 ± 0.62	29.99 ± 0.35	29.65 ± 0.36	29.65 ± 0.26	0.421	28.63 ± 0.27	29.53 ± 0.36	28.99 ± 0.28	0.293
**Positive control**	28.46 ± 0.42	29.23 ± 0.36	30.65 ± 0.26	29.46 ± 0.22	29.26 ± 0.28		27.82 ± 0.41	28.73 ± 0.25	28.67 ± 0.19	0.317
**Treatment**	29.14 ± 0.38	29.86 ± 0.19	30.72 ± 0.42	30.17 ± 0.30	29.57 ± 0.21		28.56 ± 0.23	29.93 ± 0.13	28.38 ± 0.16	0.348
** *p* **	0.537						0.196	0.326	0.307	
**Blood Mean Corpuscular Volume (MCV) Levels of Cow**										
**Control**	46.57 ± 0.38	46.72 ± 0.52	46.73 ± 0.52	47.01 ± 0.56	47.21 ± 0.33	0.473	47.13 ± 0.36	16.39 ± 0.23	16.53 ± 0.08	0.425
**Positive control**	46.88 ± 0.56	46.39 ± 0.61	46.82 ± 0.53	46.79 ± 0.35	46.96 ± 0.58		47.27 ± 0.29	16.68 ± 0.17	16.58 ± 0.09	0.507
**Treatment**	46.62 ± 0.42	46.82 ± 0.43	46.48 ± 0.62	46.72 ± 0.33	47.05 ± 0.62		47.63 ± 0.71	16.57 ± 0.34	16.65 ± 0.10	0.563
** *p* **	0.342						0.348	0.449	0.608	
**Blood Mean Corpuscular Hemoglobin (MCH) Levels of Cow**										
**Control**	17.02 ± 0.08	16.92 ± 0.13	17.98 ± 0.18	17.04 ± 0.10	16.91 ± 0.37	0.327	16.32 ± 0.18	16.39 ± 0.23	16.53 ± 0.08	0.425
**Positive control**	17.11 ± 0.04	17.08 ± 0.12	17.06 ± 0.11	17.06 ± 0.13	16.83 ± 0.21		16.42 ± 0.21	16.68 ± 0.17	16.58 ± 0.09	0.507
**Treatment**	17.08 ± 0.12	17.11 ± 0.16	17.19 ± 0.12	17.08 ± 0.11	17.06 ± 0.18		16.73 ± 0.28	16.57 ± 0.34	16.65 ± 0.10	0.563
** *p* **	0.273						0.481	0.449	0.608	
**Blood Mean Corpuscular Hemoglobin Concentration (MCHC) Levels of Cow**										
**Control**	36.28 ± 0.21	36.08 ± 0.13	36.15 ± 0.14	36.19 ± 0.19	36.02 ± 0.10	0.412	36.18 ± 0.09	36.18 ± 0.14	36.33 ± 0.12	0.642
**Positive control**	36.62 ± 0.18	36.14 ± 0.12	36.12 ± 0.15	36.08 ± 0.12	36.15 ± 0.11		35.99 ± 0.11	35.97 ± 0.09	36.28 ± 0.17	0.536
**Treatment**	36.29 ± 0.34	36.03 ± 0.13	36.07 ± 0.11	36.12 ± 0.06	36.02 ± 0.08		36.21 ± 0.28	36.21 ± 0.12	36.14 ± 0.11	0.484
** *p* **	0.328						0.268	0.298	0.300	
**Blood Platelet (PLT) Levels of Cow**										
	−60 ^b^	−45 ^a^	−30 ^a^	−15 ^a^	−7 ^a^	0.009	243.21 ± 7.63	249.34 ± 8.67	248.76 ± 6.23	0.569
**Control**	190.21 ± 6.28	198.76 ± 6.42	201.27 ± 7.38	207.68 ± 6.51	208.56 ± 5.67					
**Positive control**	191.18 ± 5.26	195.16 ± 5.67	199.37 ± 6.12	211.63 ± 7.16	211.76 ± 8.36					
**Treatment**	193.32 ± 8.34	194.53 ± 7.78	204.41 ± 8.57	206.11 ± 8.72	206.62 ± 7.63		246.72 ± 7.23	248.58 ± 7.43	248.42 ± 8.03	0.771
** *p* **	0.289						0.342	0.402	0.458	
**Blood Mean Platelet Volume (MPV) Levels of Cow**										
**Control**	5.36 ± 0.12	5.51 ± 0.10	5.41 ± 0.12	5.39 ± 0.07	5.43 ± 0.09	0.227	5.46 ± 0.06	5.42 ± 0.11	5.43 ± 0.09	0.691
**Positive control**	5.42 ± 0.10	5.47 ± 0.08	5.48 ± 0.11	5.41 ± 0.12	5.47 ± 0.08		5.61 ± 0.13	5.39 ± 0.13	5.41 ± 0.10	0.451
**Treatment**	5.59 ± 0.11	5.53 ± 0.09	5.43 ± 0.09	5.50 ± 0.09	5.51 ± 0.06		5.59 ± 0.12	5.46 ± 0.09	5.45 ± 0.08	0.598
** *p* **	0.172						0.423	0.275	0.307	
**Red Blood Cell (RBC) Levels of Cow**										
**Control**	6.67 ± 0.07	6.59 ± 0.13	6.62 ± 0.11	6.61 ± 0.06	6.63 ± 0.09	0.563	5.89 ± 0.21	5.90 ± 0.19	5.96 ± 0.20	0.295
**Positive control**	6.63 ± 0.10	6.61 ± 0.11	6.70 ± 0.14	6.67 ± 0.08	6.58 ± 0.08		5.98 ± 0.17	5.86 ± 0.17	5.93 ± 0.19	0.176
**Treatment**	6.58 ± 0.12	6.63 ± 0.08	6.65 ± 0.13	6.58 ± 0.10	6.60 ± 0.06		5.81 ± 0.09	5.93 ± 0.22	5.89 ± 0.21	0.152
** *p* **	0.192						0.502	0.573	0.624	

Values with different superscripts in the same row and column differ significantly (*p* < 0.05; *p* < 0.01). The *p* value on right side indicates significance or non-significance among the groups with respect to time, while the *p* value on the bottom indicates significance or non-significance between the groups.

**Table 3 animals-14-00150-t003:** Blood serological count values of cows fed essential oil blends before and after parturition ($\bar{x}$ ± *S*$\bar{x}$).

**Total Immunoglobulin G (IgG) Levels of Cow**							
	Before Parturition						One Day after Parturition
**Groups**	−60 ^b^	−45 ^ab^	−30 ^bc^	−15 ^c^	−7 ^ac^	*p*	
**Control**	25.19 ± 1.27	27.31 ± 1.53	30.24 ± 0.96	32.73 ± 1.58	32.16 ± 1.21 ^A^	0.001	28.36 ± 1.13
**Positive control**	25.28 ± 1.42	27.46 ± 1.28	32.28 ± 0.99	33.42 ± 0.98	32.97 ± 0.93 ^A^		28.42 ± 1.16
**Treatment**	25.13 ± 1.36	28.29 ± 1.62	32.19 ± 1.07	34.57 ± 1.19	35.73 ± 1.17 ^B^		31.09 ± 1.45
** *p* **	**0.001**						0.112
**Total Leukocyte Count (TLC) Levels of Cow**							
**Control**	6.13 ± 0.11	6.09 ± 0.13	6.11 ± 0.12	6.17 ± 0.21	6.72 ± 0.32	0.028	7.72 ± 0.21 ^A^
**Positive control**	6.01 ± 0.06	6.08 ± 0.09	6.18 ± 0.13	6.24 ± 0.12	6.84 ± 0.26		7.62 ± 0.27 ^A^
**Treatment**	6.05 ± 0.08	6.11 ± 0.10	6.16 ± 0.17	6.18 ± 0.19	6.93 ± 0.39		6.40 ± 0.22 ^B^
** *p* **	0.417						0.001
**Lymphocyte Count (LC) Levels of Cow**							
**Control**	2.51 ± 0.14	2.58 ± 0.11	2.57 ± 0.12	2.69 ± 0.16	2.99 ± 0.16	0.041	3.47 ± 0.19 ^A^
**Positive control**	2.56 ± 0.19	2.53 ± 0.09	2.59 ± 0.08	2.71 ± 0.11	2.91 ± 0.09		3.40 ± 0.13 ^A^
**Treatment**	2.49 ± 0.16	2.56 ± 0.07	2.63 ± 0.14	2.59 ± 0.12	2.96 ± 0.10		3.09 ± 0.16 ^B^
** *p* **	**0.377**						0.019
**Neutrophil Count (NC) Levels of Cow**							
**Control**	2.81 ± 0.10	2.82 ± 0.08	2.80 ± 0.06	2.77 ± 0.16	2.91 ± 0.16	0.045	4.06 ± 0.12 ^A^
**Positive control**	2.76 ± 0.06	2.79 ± 0.07	2.73 ± 0.09	2.80 ± 0.13	3.03 ± 0.27		3.89 ± 0.26 ^A^
**Treatment**	2.67 ± 0.11	2.69 ± 0.09	2.75 ± 0.12	2.71 ± 0.11	3.06 ± 0.19		3.21 ± 0.19 ^B^
** *p* **	0.656						0.001
**Monocyte Count (MC) Levels of Cow**							
**Control**	0.316 ± 0.022	0.338 ± 0.021	0.341 ± 0.027	0.347 ± 0.029	0.386 ± 0.021	0.001	0.459 ± 0.025 ^A^
**Positive control**	0.319 ± 0.019	0.341 ± 0.026	0.346 ± 0.019	0.348 ± 0.016	0.376 ± 0.019		0.465 ± 0.031 ^A^
**Treatment**	0.308 ± 0.018	0.357 ± 0.024	0.351 ± 0.023	0.354 ± 0.011	0.379 ± 0.023		0.402 ± 0.017 ^B^
** *p* **	0.173						0.042

Values with different superscripts in the same row and column differ significantly (*p* < 0.05; *p* < 0.01). The *p* value on right side indicates significance or non-significance among the groups with respect to time, while the *p* value on the bottom indicates significance or non-significance between the groups.

**Table 4 animals-14-00150-t004:** Blood biochemical parameters of cow ($\bar{x}$ ± *S*$\bar{x}$).

Parameters	One Day before Parturition				One Day after Parturition			
	Control	Positive Control	Treatment	*p*	Control	Positive Control	Treatment	*p*
Aspartate Aminotransferase (AST)	79.46 ± 1.21	79.21 ± 1.42	80.62 ± 1.36	0.267	87.63 ± 0.81	86.81 ± 0.76	86.98 ± 1.12	0.357
Alanine Aminotransferase (ALT)	21.71 ± 0.36	21.63 ± 0.39	21.47 ± 0.31	0.228	23.52 ± 0.36	23.28 ± 0.21	23.12 ± 0.32	0.463
Alkaline Phosphatase (ALP)	102.31 ± 4.51	100.29 ± 5.19	103.23 ± 3.76	0.496	99.59 ± 4.23	97.27 ± 5.12	96.62 ± 3.85	0.216
Gamma Glutamyl Transferase (GGT)	23.79 ± 1.24	22.89 ± 1.64	23.59 ± 1.71	0.259	24.32 ± 0.72	23.95 ± 0.59	24.47 ± 0.61	0.189
Non-Esterified Fatty Acid (NEFA)	0.169 ± 0.024	0.154 ± 0.032	0.163 ± 0.042	0.359	0.572 ± 0.325 ^A^	0.516 ± 0.196 ^A^	0.469 ± 0.214 ^B^	0.001
Beta Hydroxy Butyric Acid (BHBA)	0.089 ± 0.023	0.101 ± 0.019	0.92 ± 0.016	0.298	0.879 ± 0.063 ^A^	0.846 ± 0.039 ^A^	0.776 ± 0.025 ^B^	0.001
GLUCOSE	60.12 ± 2.39	58.87 ± 1.63	61.42 ± 3.14	0.472	49.98 ± 1.12	50.52 ± 1.36	52.01 ± 1.51	0.564

Values with different superscripts in the same row differ significantly (*p* < 0.01). The *p* value on the right side indicates significance between the groups.

**Table 5 animals-14-00150-t005:** Blood hematological and biochemical parameters in calves ($\bar{x}$ ± *S*$\bar{x}$).

Groups	Haemoglobin	Haematocrit	Blood MCV	Blood MCH	MCHC	Platelet Count	MPV	RBC	IgG	TLC	Lymphocyte	Neutrophil	Monocyte	AST	ALT	ALP	GGT
Control	9.72 ± 0.43	31.13 ± 0.23	32.25 ± 1.05	11.64 ± 0.34	11.64 ± 0.34	249.42 ± 2.17	5.41 ± 0.08	6.39 ± 0.21	0.42 ± 0.04	7.41 ± 0.36	3.66 ± 0.24	3.46 ± 0.13	0.419 ± 0.021	49.56 ± 0.96	12.21 ± 0.43	124.42 ± 6.23	73.65 ± 3.66
Positive Control	9.68 ± 0.21	30.81 ± 0.17	32.16 ± 1.12	11.79 ± 0.27	11.79 ± 0.27	247.71 ± 1.91	5.45 ± 0.05	6.28 ± 0.36	0.39 ± 0.03	7.49 ± 0.32	3.52 ± 0.28	3.62 ± 0.17	0.403 ± 0.016	49.45 ± 1.03	13.63 ± 0.28	120.19 ± 7.53	68.67 ± 4.39
Treatment	9.76 ± 0.27	30.96 ± 0.19	32.41 ± 0.96	11.92 ± 0.32	11.92 ± 0.32	250.16 ± 2.10	5.41 ± 0.05	6.21 ± 0.32	0.41 ± 0.02	7.19 ± 0.26	3.43 ± 0.27	3.58 ± 0.26	0.395 ± 0.027	47.61 ± 1.12	12.52 ± 0.32	119.58 ± 6.42	74.25 ± 3.92
*p*	0.746	0.853	0.837	0.694	0.694	0.392	0.876	0.718	0.252	0.176	0.203	0.314	0.197	0.319	0.256	0.148	0.148

## Data Availability

All the data obtained during this study is contained within this article. No new data, other than presented in the current article, is created.

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
