# Peer review of "Effect of a Dietary Essential Oil Blend in Dairy Cows during the Dry and Transition Period on Blood and Metabolic Parameters of Dams and Their Calves"

_animals, 2024, doi:10.3390/ani14010150_

Round 1

Reviewer 1 Report

Comments and Suggestions for Authors

Revision letter

Animals

Effect of addition of essential oil blend in dairy cow’s ration during dry period on immune system and some metabolic parameters of dame and calf

The study is aimed at testing the effects of dietary dry period of a supplement containing herbal plant essential oils, on some hematological, biochemical, and immunological parameters of cow and their calves.

The manuscript is interesting but major revisions are needed. It it is not clear why postpartum and peripartum data are evaluated in separate blocks.

The discussion is not very readable and should be improved. There are to many tables and they do not stand alone from the text. In the tables acronyms and units of measurement are missing

Detailed comments:

suggested title Effect of essential oil blend supplementation during dry and transition period of dairy cow on some hematological, biochemical, metabolic and immunological parameters of dames and calves

Abstract: the aim is missing, please add it to the abstract

Introduction

L 101-102 Rephrase the purpose to make it better fit with the title. It could be “The study is aimed at the effects of supplementation of essential oil blend (Digestarom DairyTM) in dairy cow’s ration during dry period on some hematological, biochemical, metabolic and immunological parameters of of dames and calves”

L 103-119 please, delete

Materials and methods

L 129 please, specify how long the supplementation has been given

L 146-147 check the sampling times. They are different with what reported at l 17 of the abstract and tables

L 154 and 178 result and datas on milk colostrum and milk are not reported

172-173 it is unclear why postpartum and peripartum data are evaluated in separate blocks

L 174 and 178 – the sentences are redundant, shorten and put together in order to improve readability

Results and discussion

L 251-254 the acronyms are already described in materials and methods; add the reference to the table

Line 252 “ PLT “in contrast with table, please check

L 254-259 it is not clear what results this part refers to

L 272 and 275: repetition

L 277 “ when data were evaluated “ not clear

L 260-267; 302-307; 317-324 this parts seem more like a list of articles, it should be rewritten in a more organic way

L 283 and  291 add the reference to the table

337-338 please rephrase

L 313-316 please clarify

L 341 NED: not clear

Conclusions

L 354 “ Digestarom” change with “Digestarom TM

L 355 “according the statistical evaluations of the data” this sentence is redundant

L 358 “after birth” change with after “delivery”

The number of tables (15) should be reduced for example the tables could be 4: 2 for the blood, biochemical and immunological parameters of cows and 2 for the parameters of calves.

Author Response

Dear Reviewer, Thank you very much for detailed review of our manuscript to improve it. We have taken your suggestions and extensively revised the manuscript. The detailed point to point response is attached in the file. 

Reviewer 2 Report

Comments and Suggestions for Authors

General remarks:

The topic of this manuscript is interesting for Animals’ readers, I find some serious problematic parts in the text.

Detailed opinions:

Recommended title:

Effect of herbal plants additives in dairy cow’s diet during transition period on some blood and metabolic parameters of cows and their calves

Introduction

Lines 36-39 and lines 43-45: these sentences are the same! Please modify the text, and clarify it!

lines 101-104: please rewrite this paragraph!

lines 105-119: delete these paragraphs!

Please add the aims of this study!

Materials and Methods

line 126: positive control? What does it mean?

lines 128-130: when beginning and how long the study and feeding the Digestarom?

Please explain the Digestarom characteristics (and please add more information about Digestarom into the Introduction section)!

line 152: what parameters were analysed by the Mindray device?

line 153: How separated the serum and plasma? Please add centrifuge data!

line 155: if use EDTA and native tubes, please it indicates on “a” and “b” points!

line 166: please add the name of the device!

lines 170-171: logarithmic transformation is not clear! Please clarify it!

Why did not use two-way ANOVA such as GLM (factors: groups and times)?

Results

Tables: readability of all tables is weak. I recommend all Tables rewrite according to factors (group + time), and I think it is better if the calves’ results are separate from cows’ results!

lines 260-263: this is a contradiction: “Kumar and Pachauri (2000) reported that blood hemoglobin levels decreased during the dry period of cows fed in pasture. Similar to this study, Baqi and Rahman (1987) and Rajora and Pachauri (1994) reported that the blood hemoglobin level increased as the gestation period progressed.“ Better if use: In contrast or similar phrase!

lines 269-271: previously no available data about housing technology parameters of cows! How to reduce environmental stress?

lines 276-279: this paragraph is not clear! Please rewrite it! And change the mother word to cow (everywhere in the text)!

lines 282-284: “gradual increase…” among time of pregnancy? Please clarify it!

line 341: NED: please add full name for the first time!

Conclusions

line 353: “volatile fatty acid mixture”

lines 353-355: this sentence is not clear, please rewrite and focus on the experiment (cows, calves)!

lines 355-358: this sentence is not clear, too! Please define which animals’ parameters were improved by feeding Digestarom!

I think this section must be rewritten, with regard to results!

Author Response

(The authors gave the same response as above.)

Reviewer 3 Report

Comments and Suggestions for Authors

This study compared the use of Levamisole and essential oils on blood parameters in cows an their offspring. The second half of the introduction gives some good background to previous literature on essential oils, which is helpful as many readers are likely to be unfamiliar with them. Some additional information is needed in the introduction on use of Levamisole - why did you use this as a control group? This is vital as at no point do you explain its use as a control group.

The English needs substantial reviewing throughout the paper to make it of publishable standard. The results/ discussion section is difficult to follow and needs substantial re-working to aid in clarity and readability.

Specific comments on each section are given below:-

- Title typo - dam not dame

 -  Line 11: the abstract needs an opening sentence to tell the reader what the study is about.

 -  Line 23: extra space present following 7th

-Line 45: repetition of economic losses - can you give an example of how much is lost?

-Line 48 - colostrogenesis happens pre-calving

-Line 74: might be worth defining carminative and colororetic

-Line 104: delete this text which remains from the template.

-Line 123: avoid starting a sentence with a number is possible.

-Line 123: use the past tense (had not have)

-Line 127: as mentioned, explanation on the use of Levamisole is needed

-Line 128: what essential oils were in this supplement? Needs specific information about the product.

-Line 130 - where were the animals from? Yield? Age? Why using 2nd lactation animals? sample size calculation?

-Line 132: the full name should not be in the brackets, the abbreviation should be

-Line 187: table formatting not correct

-Tables all need definitions for the letters putting into them. I would select the most relevant tables to put in the main text, and others can go in the supplementary material. Currently there are too many tables, and they need referring to within the body of the results text.

-Line 254 - extra space after stages

-Line 260 to 266: this is just a list of of findings from other literature, and needs rephrasing to improve readability.

-Line 287: should present P<0.001, not just P=0.000

-Line 312: can you give any explanation as to why the IgG level might have been higher in the group fed the essential oils?

Comments on the Quality of English Language

poor throughout

Author Response

(The authors gave the same response as above.)

Round 2

Reviewer 2 Report

Comments and Suggestions for Authors

I carefully evaluated this revised version of the manuscript, all sections were improved by the authors according to the my comments, so I recommend this manuscript for publishing in the Animals journal in the present form!

Author Response

Dear Reviewer,

Thank you so much for your time and efforts to improve our article. 

Reviewer 3 Report

Comments and Suggestions for Authors

The authors have made substantial revisions to the paper. However there are still many English grammatical errors throughout, some of which are specific below, and others that are not. The paper should be proof read for grammatical correctness. There are also some small additions needed to the discussion to actually tell the reader the direction of significant changes detected so they don't have to go back and find it in the tables. Some interpretation to the biological relevance in the level of the changes of the parameters is needed for interpretation.

The title is still not correct - it could be changed to something like: effect of a dietary essential oil blend for dairy cows during the dry and transition periods on some blood and metabolic parameters of dams and their calves

Line 15 - cows face, not cow faces

Line 17 - remove the , after blend

Line 43 - The transition period

Line 114 - what is DSM? if this is the company, this needs putting in brackets with the country of origin after the product name.

Line 139 - repetition of experimental followed by experiment. change the second to study. Was a sample size calculation done?

Line 151 - misspelled Levamisole.

The tables need to have the letters for the groups defined in the table headers.

Line 350 to 352 - what were these changes, what was the significant difference, and is this difference level biologically relevant?

Same comment with 369 - is this biologically relevant?

Line 386 - what was the difference, higher or lower after parturition? Did the essential oil make the food more palatable, could this account for the NEFA/ BHB finding?

Comments on the Quality of English Language

Improvements have been made, but minor corrections are still needed

Author Response

Dear Reviewer,

Thank you for your time and effort in reviewing it again. We have tried to incorporate all the corrections as suggested by you. We have removed grammar issues. Additions in the discussion section have been added for better readability of the text. To make it clear regarding the significant changes, we have also mentioned either it is significantly increased or decreased in the discussion section. We have also clarified which group values are changing. Justification about relevance is also added. All the corrections have been made in the main text file of the article. 

Point to Point response is given in the attached response file. 
